

# Methane emissions due to reservoir flushing: a significant emission pathway?

Ole Lessmann[1], Jorge Encinas Fernández[1], Karla Martínez-Cruz[1], Frank Peeters[1]

[1]Department of Biology, Limnological Institute, University of Konstanz, Konstanz, 78464, Germany

*Correspondence to*: Ole Lessmann (ole.lessmann@uni-konstanz.de); Frank Peeters (frank.peeters@uni-konstanz.de)

**Abstract.** Reservoirs can emit substantial amounts of the greenhouse gas methane ($CH_4$) via different emission pathways. In some reservoirs, reservoir flushing is employed as a sediment management strategy to counteract growing sediment deposits that threaten reservoir capacity. Reservoir flushing utilizes the eroding force of water currents during water level drawdown to mobilize and transport sediment deposits through the dam outlet into the downstream river. During this process, $CH_4$ that

is stored in the sediment can be released into the water and degas to the atmosphere resulting in $CH_4$ emissions. Here, we assess the significance of this $CH_4$ emission pathway and compare it to other $CH_4$ emission pathways from reservoirs. We measured seasonal and spatial $CH_4$ concentrations in the sediment of Schwarzenbach Reservoir, providing one of the largest datasets on $CH_4$ pore water concentrations in freshwater systems. Based on this dataset we determined $CH_4$ fluxes from the sediment and estimated potential $CH_4$ emissions due to reservoir flushing. $CH_4$ emissions due to one flushing operation can

constitute 7–14% of the typical annual $CH_4$ emissions from Schwarzenbach Reservoir, whereby the amount of released $CH_4$ depends on the timing of the flushing operation within the season. The larger the thickness of the sediment layer mobilized during the flushing operation the larger the average $CH_4$ concentration per unit volume of flushed sediment. This suggests that regular flushing of smaller sediment layers releases less $CH_4$ than removal of the same sediment volume in fewer flushing events of thicker sediment layers. In other reservoirs with higher sediment loadings than Schwarzenbach Reservoir,

reservoir flushing could cause substantial $CH_4$ emissions, especially when flushing operations are conducted frequently. Therefore, $CH_4$ emissions due to reservoir flushing must be included in estimates of annual overall greenhouse gas emissions from reservoirs that are subject to regular flushing operations.

## 1 Introduction

Worldwide ~16.7 million reservoirs exist (Lehner et al., 2011), and their number is projected to increase substantially in the

near future (Zarfl et al., 2015) because of rising demand for hydropower. Besides being used for electricity generation and energy storage, reservoirs can serve multiple purposes such as water supply, flood control, irrigation and navigation (WCD (World Commission on Dams), 2000). In the past, hydropower was widely considered a greenhouse gas (GHG) neutral form of energy (Hoffert et al., 1998). Today, we know that the required reservoirs represent a significant source of GHG emissions, especially of the potent GHG methane ($CH_4$), and researchers have estimated that reservoirs contribute around



17.7–70.0 Tg $CH_4$ to the annual global budget of atmospheric $CH_4$ (St. Louis et al., 2000; Bastviken et al., 2011; Deemer et al., 2016; Rosentreter et al., 2021).

      $CH_4$ can be emitted from reservoirs via different pathways such as ebullition, plant-mediated transport, diffusion across the water-atmosphere interface, degassing during turbination ("drawdown flux") and during spring or fall turnover as storage flux (Bastviken et al., 2011). $CH_4$ is typically produced in the anoxic part of the sediment (Le Mer and Roger, 2001).

At oxic interfaces within the sediment, at the sediment surface or within the water column, $CH_4$ is at least partly oxidized by methane-oxidizing bacteria (Bastviken et al., 2002) before it reaches the water surface from where it is emitted as diffusive flux to the atmosphere. $CH_4$ flux via ebullition and plant-mediation, on the other hand, can bypass oxic interfaces, thus avoiding oxidation (Chanton and Whiting, 1995). In anoxic deep water of lakes and reservoirs, $CH_4$ typically accumulates and can reach large concentrations of stored $CH_4$ (Ragg et al., 2021). Intensive vertical mixing during spring and fall

overturn causes rapid transport of the stored $CH_4$ to the water surface from where it diffuses to the atmosphere. This storage flux can substantially exceed the annual diffusive emissions during stratified conditions (Encinas Fernández et al., 2014).

      While most of these emission pathways usually exist in both lakes and reservoirs, the drawdown flux of $CH_4$, i.e., the degassing of $CH_4$ during turbination, occurs only in reservoirs where it can become the dominant source of $CH_4$ emissions (Kemenes et al., 2007), especially if the turbinated water is drawn from anoxic deep water where large amounts of

$CH_4$ are stored. Ebullition of $CH_4$ is often observed as the dominant emission pathway, especially in shallow reservoirs (DelSontro et al., 2010; Sobek et al., 2012). It is well known that changes in hydrostatic pressure can induce bubble formation and release from sediments (Maeck et al., 2014; Harrison et al., 2017; Encinas Fernández et al., 2020). Therefore, water level drawdowns in reservoirs can substantially increase the ebullition flux of $CH_4$ due to decreasing pressure, which can become particularly large during fall drawdown or occur regularly during diel pumped-storage operations (Harrison et

al., 2017; Encinas Fernández et al., 2020).

      Growing sediment deposits in reservoirs, caused by particles introduced from the catchment and organic matter produced within the system, pose a challenge to maintaining reservoir capacity (e.g., by decreasing reservoir storage volume). Globally, reservoir storage capacity is decreasing and is estimated to be completely lost for most reservoirs within 200–300 years if sediment management strategies are not adopted (ICOLD, 2009). Sediment management strategies are

applied in reservoirs to reduce sediment yield, route sediments or remove already deposited sediment (Morris, 2020; Petkovšek et al., 2020). Deposited sediment can be removed by mechanical removal (e.g., dredging) or hydraulic flushing, which utilizes the eroding force of water currents during water level drawdown. The sediment is then mobilized in the water and transported through a dam outlet to the downstream river section. Consequently, any $CH_4$ previously stored in the sediment pore water is also mobilized into the water stream and can eventually degas to the atmosphere at the reservoir

surface, during turbination, or downstream of the reservoir, leading to $CH_4$ emissions. In order to assess the relative importance of $CH_4$ emissions due to reservoir flushing compared to other typical $CH_4$ emission pathways, it is important to consider reservoir flushing frequency and sediment flushing volume. Among reservoirs with relatively regular flushing operations, flushing frequencies between twice a year and once every five years have been reported (Brandt and Swenning,



1999; Chang et al., 2003; Kantoush and Sumi, 2010; Fruchard and Camenen, 2012; Grimardias et al., 2017; Sumi et al.,
2017; Antoine et al., 2020), but there are also reservoirs that are flushed irregularly (Sumi et al., 2017). Reservoirs with
higher sedimentation rates need to be flushed more frequently than others. For instance, the Tapu reservoir in Taiwan is
characterized by high sedimentation rates induced by heavy rainfalls and steep-sloped mountains and was flushed 10 times
between 1991 and 1996 (Chang et al., 2003). The success of the reservoir flushing operation can be measured by the volume
of sediment removed from the reservoir, which depends on reservoir geometry, sediment characteristics and operation
strategy (Morris, 2020; Petkovšek et al., 2020).

In this study, we investigated $CH_4$ emissions due to reservoir flushing, a pathway that has not yet been included in
estimates of $CH_4$ emissions from reservoirs. We determined the amount of $CH_4$ stored in the sediment pore water of a
reservoir at different seasons and estimated potential $CH_4$ emissions resulting from reservoir flushing scenarios.
Furthermore, we assessed the relative importance of reservoir flushing in comparison to other typical $CH_4$ emission pathways
in reservoirs.

## 2 Material and methods

### 2.1 Study Site

Field measurements were conducted in Schwarzenbach Reservoir, which is located in the northern part of the Black Forest in
southwest Germany (48°39.334′N, 8°19.630′E) at ~660 m above sea level (a.s.l.). The reservoir is operated as a pumped-
storage hydroelectric energy system. At a maximum storage capacity of $14.4 \times 10^6$ m³, the reservoir is 47 m deep and covers
a surface area of ~0.66 km². The reservoir's alkalinity and salinity are typically in the order of 0.3 $mmol_{eq}$ L⁻¹ and 0.04 g kg⁻¹,
respectively. The reservoir receives input from two natural creeks (Schwarzenbach and Seebach) located at the western end
of the basin and from an artificial channel in the south (Raummuenzachstollen) that collects water from the immediate
catchment area (Fig. 1a). Due to a sediment trap located upstream of the artificial channel, the Seebach and Schwarzenbach
are the only sources for sediment transport into the reservoir. For the pumped-storage operation, an in- and outlet is based
near the dam at ~5 m above ground. The contributions of the individual inputs are described in Mouris et al. (2018).



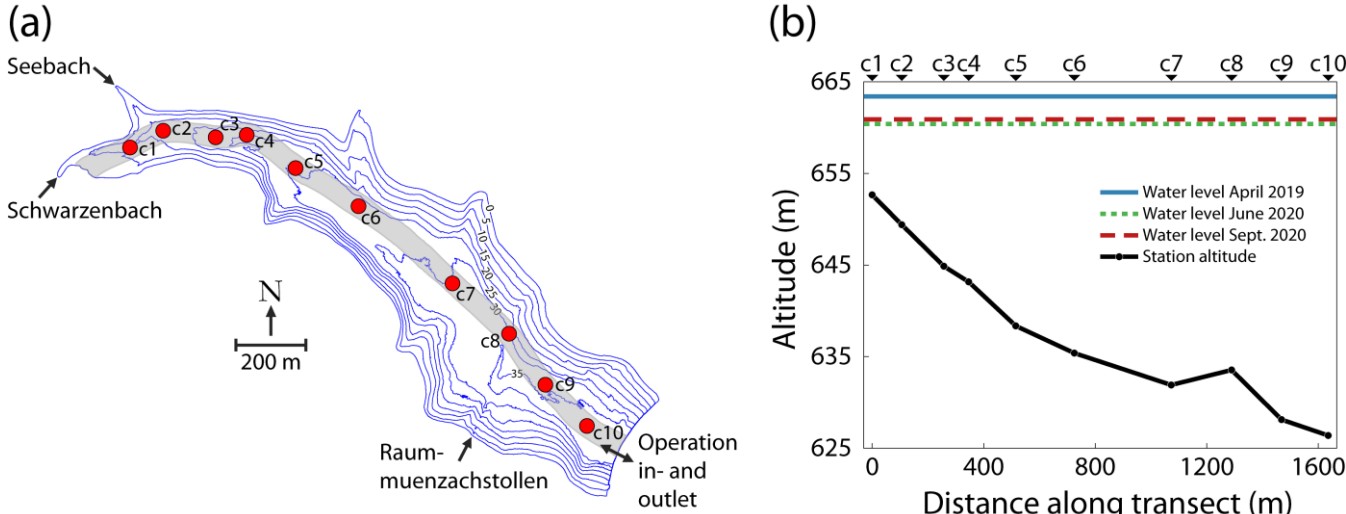

**Figure 1. (a) Bathymetric map of Schwarzenbach Reservoir showing in- and outflows and the position of the sampling stations (c1 – c10). The grey shaded area symbolizes the estimated basin-wide erosion channel with a 60 m width. (b) Sediment altitude at the sampling stations and water level during three field campaigns (April 2019, June 2020, and September 2020). The individual sampling stations are indicated above the panel.**

## 2.2 Field measurements

Three field campaigns (April 2019, June 2020 and September 2020) were conducted at Schwarzenbach Reservoir, during which a total of 47 sediment cores were retrieved, i.e., 13 in April 2019 and 17 in each of the campaigns in June and September. The sediment cores were sampled along the transect close to the thalweg (Fig. 1a). At each sampling station, two sediment cores were retrieved, except for station c5 in April 2019 and June 2020, and station c3 in September 2020, where only one sediment core was retrieved. The transect length was ~1.6 km, and the altitude of the sediment cores ranged between 626.4 and 652.7 m (Fig. 1b). In April 2019, the reservoir's water level was around 3 m higher than in June and September 2020. In April 2019, sediment cores were taken at station c1, c2, c4, c5, c6, c7 and c9 and in June and September 2020 at station c1, c2, c3, c5, c6, c7, c8, c9 and c10. Vertical temperature and dissolved oxygen (DO) profiles were measured in the water column at the respective stations with a multiparameter probe (CTD probe, RBR Ltd., Ottawa, Canada, equipped with a temperature and oxygen optode RBRcoda T.ODO fast). The temperature of the water overlaying the sediment (~0.5 m above the sediment surface) served as a proxy for the sediment temperature. Water samples for $CH_4$ analysis were taken using a 2-L water sampler (Limnos, Finland).

## 2.3. Measurements of $CH_4$ concentrations

Sediment cores were taken with a gravity corer equipped with a PVC liner of 600 mm length and an inner diameter of 58 mm. The liner was capped using a rubber stopper or a liner cap. Within 1–3 hours after sampling, the sediment cores were processed at the nearest shore. Sediment subsamples were taken through 0.6 cm pre-drilled holes (1 cm vertical spacing) in the PVC liner using 1 ml cutoff syringes at sediment depths of: 0.25, 1.25, 2.25, 3.25, 4.25, 5.25, 7.25, 10.25 and 15.25 cm.





Each sediment subsample was immediately inserted into a 100 ml glass bottle (DWK Life Science GmbH, Germany). The glass bottles were filled completely with demineralized water and closed using a PTFE-coated silicone septum (DWK Life Science GmbH, Germany). The bottle was shaken vigorously, enabling the pore water to dissolve into the water. The procedure to measure the $CH_4$ concentration in the water was adapted from Hofmann et al. (2010). Briefly, 50 ml liquid was sampled with a 50 ml syringe sediment settling and injected into a 100 ml glass injection vial (DWK Life Science GmbH,

Germany) containing, 20–30 g of NaCl ( ≥ 99,5 %, p.a., ACS, ISO, Carl Roth GmbH, Germany) and 30–40 ml of demineralized water. After sample injection, the $CH_4$ degassed into the headspace (~35 ml) due to the oversaturated NaCl solution. The bottles were stored upside down until further processing to prevent $CH_4$ loss over time through the septum. The $CH_4$ concentration in the equilibrated headspace of the injection vial was measured using a gas chromatograph equipped with a flame ionization detector (GC 6000, Carlo Erba Instruments, UK). The $CH_4$ concentration in the water sample was

obtained by referring the measured $CH_4$ concentration in the headspace to the volume of the respective water sample. With the concentration of $CH_4$ in the water sample $C_{CH4,ws}$, the volume of the glass bottle $V_{gb}$, the volume of the sediment sample $V_{ss}$, the volume of the pore water $V_{pw}$ and the porosity $\phi$, the concentration of $CH_4$ per sediment volume $C_{CH4,sed}$ and the concentration of $CH_4$ in the pore water $C_{CH4,pw}$ were calculated as:

$$C_{CH4,sed} = \frac{C_{CH4,ws}*(V_{gb}-(V_{ss}-V_{pw}))}{V_{ss}} \qquad (\text{mmol L}^{-1}) \qquad (1)$$

$$C_{CH4,pw} = \frac{C_{CH4,sed}}{\phi} \qquad (\text{mmol L}^{-1}) \qquad (2).$$

The porosity of the sediment describes the ratio of void volume that is occupied by the pore water, to the total volume (Brimhall and Dietrich, 1987) and was calculated for each sediment subsample accordingly:

$$\phi = \frac{V_{pw}}{V_{ss}} \qquad (\text{-}) \qquad (3).$$

However, because the difference between $V_{ss}$ and $V_{pw}$ was small, the uncertainty of the volume ratios determined from

weight measurements was rather large for individual sediment samples from specific depth and location. Therefore, we determined the average vertical profile of porosity in the sediment by fitting a cubic function of depth to all porosity profiles from all sampling campaigns (Fig. S1). The porosities of this average porosity profile were used to calculate the pore water concentrations $C_{CH4,pw}$. Vertical profiles of $C_{CH4,pw}$ were obtained by measuring at sediment depths between 0.25 and 15.25 cm. Missing data were estimated by bilinear interpolation of the $CH_4$ distribution. Missing values at boundaries were

estimated by extrapolation, assuming constant concentrations below the depth of the deepest available measurement. Inter- and extrapolated values constituted ~10% of all data points.

## 2.4. Diffusive CH₄ flux from the sediment into the water column

Assuming molecular diffusion within the sediment and using Fick's first law of diffusion accounting for the porosity and the tortuosity of the sediment, the vertical flux of $CH_4$ in the sediment is given by Berner (1980):

$$F_{sed} = -\phi(D_{CH4}\,\theta^{-2})\frac{\partial C}{\partial z} \qquad (\text{mmol m}^{-2}\,\text{d}^{-1}) \qquad (4)$$



where $D_{CH4}$ is the molecular diffusivity of $CH_4$ in water ($m^2$ $d^{-1}$), $\phi$ is the porosity (-), $\theta$ is the tortuosity (-), and $\partial C/\partial z$ is the vertical gradient of the $CH_4$ pore water concentration (mmol $m^{-3}$). The diffusive flux of $CH_4$ at the sediment-water interface, $F_{sed}$ (mmol $m^{-2}$ $d^{-1}$), was assumed to correspond to the diffusive flux of $CH_4$ in the uppermost part of the sediment (see also Berner 1980) and was determined using the gradient of the $CH_4$ pore water concentration obtained by linear regression of the

$C_{CH4,pw}$ from 0.25 cm, 1.25 cm, and 2.25 cm depth in the respective sediment core. $D_{CH4}$ was calculated from the Schmidt number of $CH_4$ (Wanninkhof, 1992) and the viscosity of the water (Weast, 1988), taking temperature and salinity into account. We assumed the sediment's temperature and salinity were the same as in the water overlaying the sediment. As porosity, we used the median of the porosity measurements in the upper 2.25 cm of all sediment cores ($\phi$=0.97). The tortuosity was determined using the porosity according to Boudreau (1997):

$\theta^2 = 1 - \ln(\phi^2)$        (-)                                                                           (5).

### 2.5. Stored CH₄ in the potentially eroded sediment volume due to reservoir flushing

For the estimation of the $CH_4$ storage in the sediment, missing values in the measured profiles of $C_{CH4,sed}$ were estimated by inter- and extrapolation using the same procedure as in the case of $C_{CH4,pw}$. Additionally, profiles of $C_{CH4,sed}$ were extended to 100 cm sediment depth assuming that $C_{CH4,sed}$ below 15 cm sediment depth is constant. Finally, the profiles of $C_{CH4,sed}$ were

linearly interpolated to obtain a regular 1 cm vertical resolution starting at 0.5 cm sediment depth. From these profiles, the total amount of $CH_4$ stored at different depths within the sediments of Schwarzenbach Reservoir was calculated by lateral and vertical integration.

        Based on reported channel widths during reservoir flushing from a similar-sized reservoir (Kantoush et al., 2010), the eroded sediment surface area was assumed to correspond to a 60 m wide flushing channel along the thalweg of

Schwarzenbach Reservoir (Fig. 1a). The total surface area covered by this channel was about $1.1 \times 10^5$ $m^2$. We assumed that the $CH_4$ profile from a sampling station represents the $CH_4$ concentrations also in the sediment in the proximity of the sampling station. Respective sediment areas were calculated by extending laterally to the half distance of bordering stations. For the outermost stations, sediment areas extended to the respective end of the basin. With all stations representing complementary parts of the total sediment surface area, $C_{CH4,sed}$ profiles measured at the different stations and their

corresponding sediment areas were used to calculate the amount of $CH_4$ stored per unit depth at different sediment depths across the entire channel, $S_{CH4}$ (mmol $m^{-1}$). The average concentration and the total amount of $CH_4$ in the potentially eroded sediment volume due to reservoir flushing ($C_{CH4,FSL}$ (mol $m^{-3}$) and $N_{CH4,FSL}$ (mol), respectively) depends on the thickness of the flushed sediment layer. $N_{CH4,FSL}$ was estimated from the stored methane $S_{CH4}$ by integrating vertically down to the depth of sediment erosion during flushing ($Z_e$):

$N_{CH4,FSL}(Z_e) = \int_0^{Z_e} S_{CH4}(z')dz'$     (mol)                                                           (6).

$C_{CH4,FSL}$ was obtained by dividing $N_{CH4,FSL}$ with the volume of mobilized sediment.





The relevance of $N_{CH4,FSL}$ for overall $CH_4$ emissions from Schwarzenbach Reservoir was assessed by comparing the $N_{CH4,FSL}$ stored in the potentially eroded sediment volume during a flushing event with other pathways of $CH_4$ emission from Schwarzenbach Reservoir (ebullition, diffusive $CH_4$ emissions from the reservoir surface, and degassing during turbination).

## 3 Results

### 3.1 Reservoir characterization

Schwarzenbach Reservoir is a pumped-storage system, and reservoir management can substantially change the water level. In April 2019, water levels were about 3 m higher than in June and September 2020 (Fig. 1b). The water column of Schwarzenbach Reservoir was stably stratified during all three campaigns (Fig. 2a). In June and September 2020, DO concentrations were oversaturated near the water surface and above 2 mg $L^{-1}$ throughout the entire water column (Fig. 2b). In April 2019 the DO sensor failed, but DO concentrations can be expected to be around 10 mg $L^{-1}$ as is indicated by a DO profile measured in April 2018. (Fig. 2b).

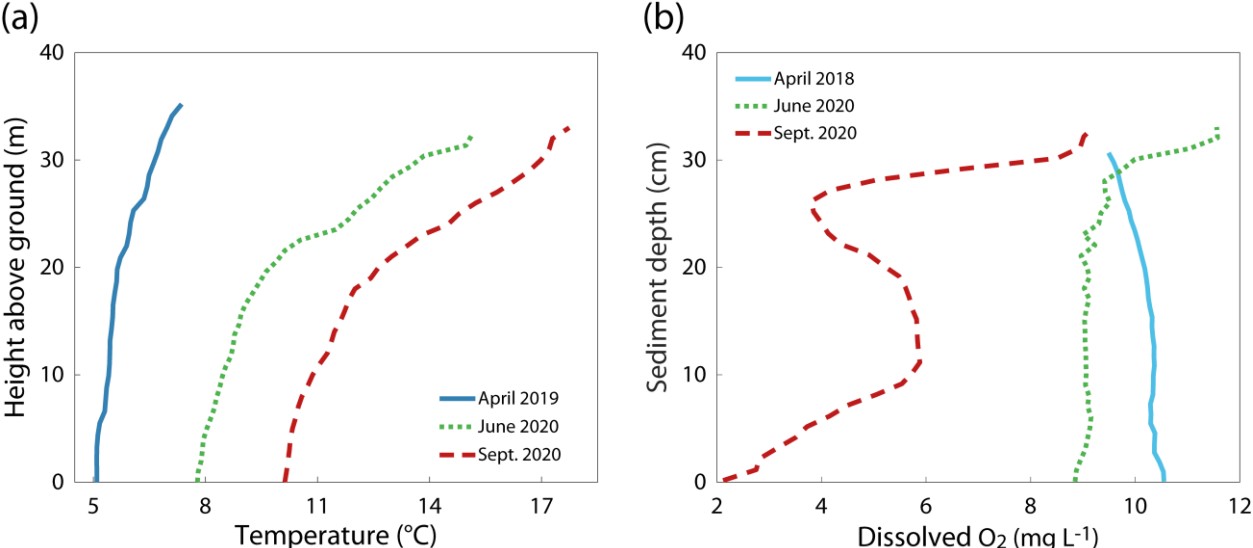

**Figure 2. Profiles of (a) temperature and (b) dissolved oxygen at station c9 during different seasons in Schwarzenbach Reservoir.**

### 3.2 Spatiotemporal dynamics of $CH_4$ in sediment pore water

The $CH_4$ concentrations in the pore water of the sediment of the reservoir differed spatially and seasonally (Fig. 3). In April 2019 and June 2020, regions with particularly large $CH_4$ concentrations were located in the center of the transect at deep sediment layers, whereas at the same depth within the sediment, $CH_4$ concentrations near the dam and the western end of the basin were considerably lower (Fig. 3a–b). In September 2020, the largest $CH_4$ concentrations were still found in the middle of the transect but were, in general more evenly distributed across the transect and larger at shallower depth than in June 2020 (Fig. 3c). For each sampling campaign, we determined a median $CH_4$ pore water profile by compiling the medians of





all measurements from the same sediment depth into a vertical profile (Fig. 3d). With the progressing season, we observed an increase in median $CH_4$ pore water concentrations and stronger vertical $CH_4$ gradients in the upmost 5 cm of the sediment. In April 2019, the median $CH_4$ concentration profile was characterized by relatively low concentrations and a weak vertical

gradient in the upper sediment layers, followed by an almost linear increase in $CH_4$ concentrations. In contrast, the median $CH_4$ profiles in June and September 2020 both showed stronger gradients in the upper sediment layers with a more saturated curve in deeper layers. However, in September 2020, the overall $CH_4$ concentrations were much larger in all but the deepest sampled sediment layers. The mean and median $CH_4$ concentration across all measurements in April 2019 were 0.27 mmol $L^{-1}$ and 0.16 mmol $L^{-1}$, in June 2020 0.37 mmol $L^{-1}$ and 0.34 mmol $L^{-1}$, and in September 2020 0.63 mmol $L^{-1}$ and 0.68

mmol $L^{-1}$, respectively, indicating an increase of overall $CH_4$ concentration in the pore water with the progressing season.

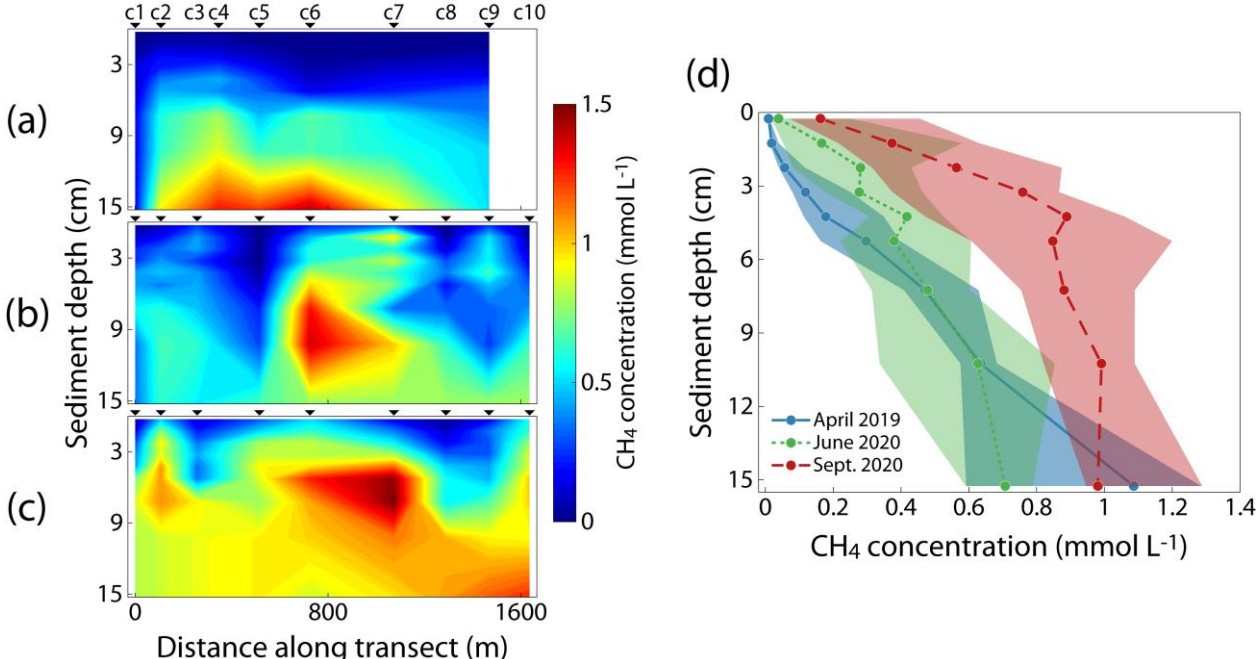

**Figure 3. (a–c) Spatial distribution of $CH_4$ pore water concentration in the top 15 cm of the sediment along the transect in April 2019 (a), June 2020 (b) and September 2020 (c). The individual sampling stations are indicated above the panels. d) Median $CH_4$**
**concentration of all pore water profiles within each campaign. Shaded areas display the 25th and 75th percentile.**

### 3.3 $CH_4$ storage estimation

To estimate the stored $CH_4$ that is potentially emitted due to sediment erosion during reservoir flushing of Schwarzenbach Reservoir, we calculated $N_{CH4,FSL}$, the amount of $CH_4$ stored between the top of the sediment and a hypothetical erosion depth within the 60 m wide erosion channel centered around the thalweg along the basin (Fig. 4, the channel is indicated in

Fig. 1a). With the progressing season, $N_{CH4,FSL}$ was consistently higher at all erosion depths. Assuming an erosion depth of 15 cm, $N_{CH4,FSL}$ was 6.5, 8.1 and 12.5 kmol $CH_4$ in April 2019, June 2020 and September 2020, respectively.



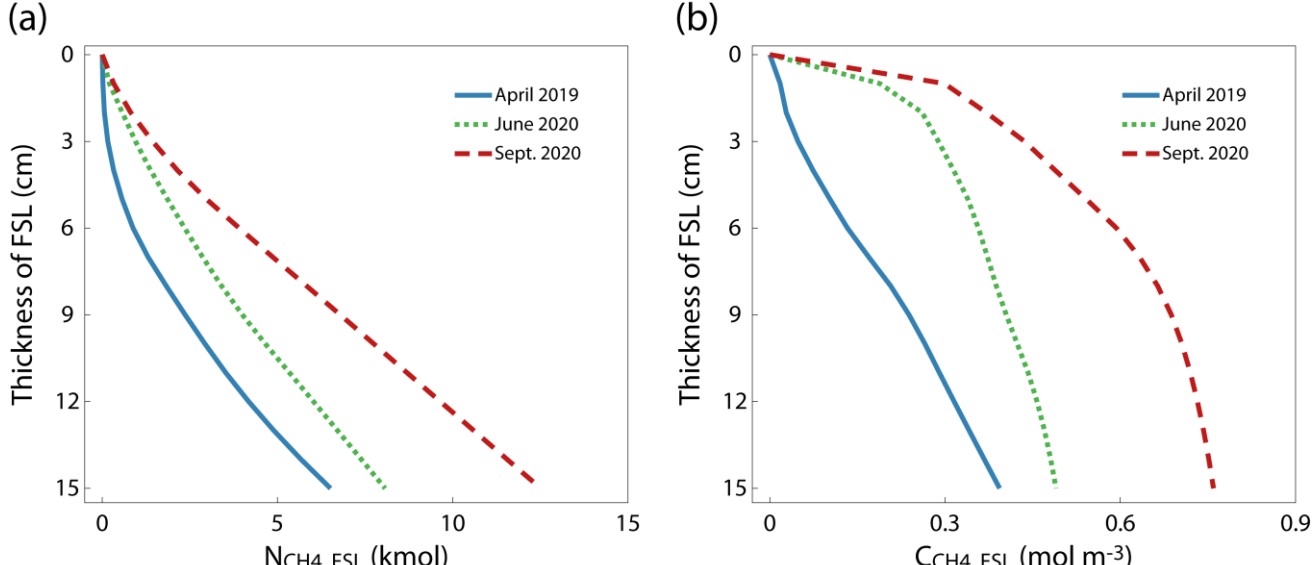

**Figure 4. (a) Stored amount of CH$_4$ in the flushed sediment layer, $N_{CH4,SFL}$, as a function of layer thickness. The thickness of the flushed sediment layer corresponds to the erosion depth of the sediment in the 60 m wide erosion channel. (b) The average CH$_4$ concentration in the flushed sediment layer.**

### 3.4 CH$_4$ fluxes at the sediment-water interface

Diffusive CH$_4$ fluxes at the sediment-water interface $F_{sed}$ were determined for each campaign at each station along the transect (Fig. 5c). The overall variability of $F_{sed}$ from all stations measured during a campaign increased with the progressing season from April to September (Fig. 5a). Compared to the range of $F_{sed}$ in April 2019, the range of $F_{sed}$ was ~3.5 times larger in June 2020 and ~4 times larger in September 2020. While the lowest measured $F_{sed}$ were very similar across the three campaigns (between 0.02 and 0.06 mmol m$^{-2}$ d$^{-1}$), the median and maximum CH$_4$ fluxes increased from April to September. The average CH$_4$ flux over all measurements was 1.02 mmol m$^{-2}$ d$^{-1}$. Sediment temperatures, approximated by water temperature ~0.5 m above the sediment, increased with the season (Fig. 5b). In April 2019, the temperatures along the transect were rather uniform, ranging between 5.1 °C (c9) and 5.9 °C (c1). Temperatures were higher in June 2020, ranging from 7.7 °C (c10) to 12.3 °C (c1) and in September 2020, ranging between 10.1 °C (c9) and 16.0 °C (c1). In June and September 2020, temperatures were more elevated in the shallower part of the basin.





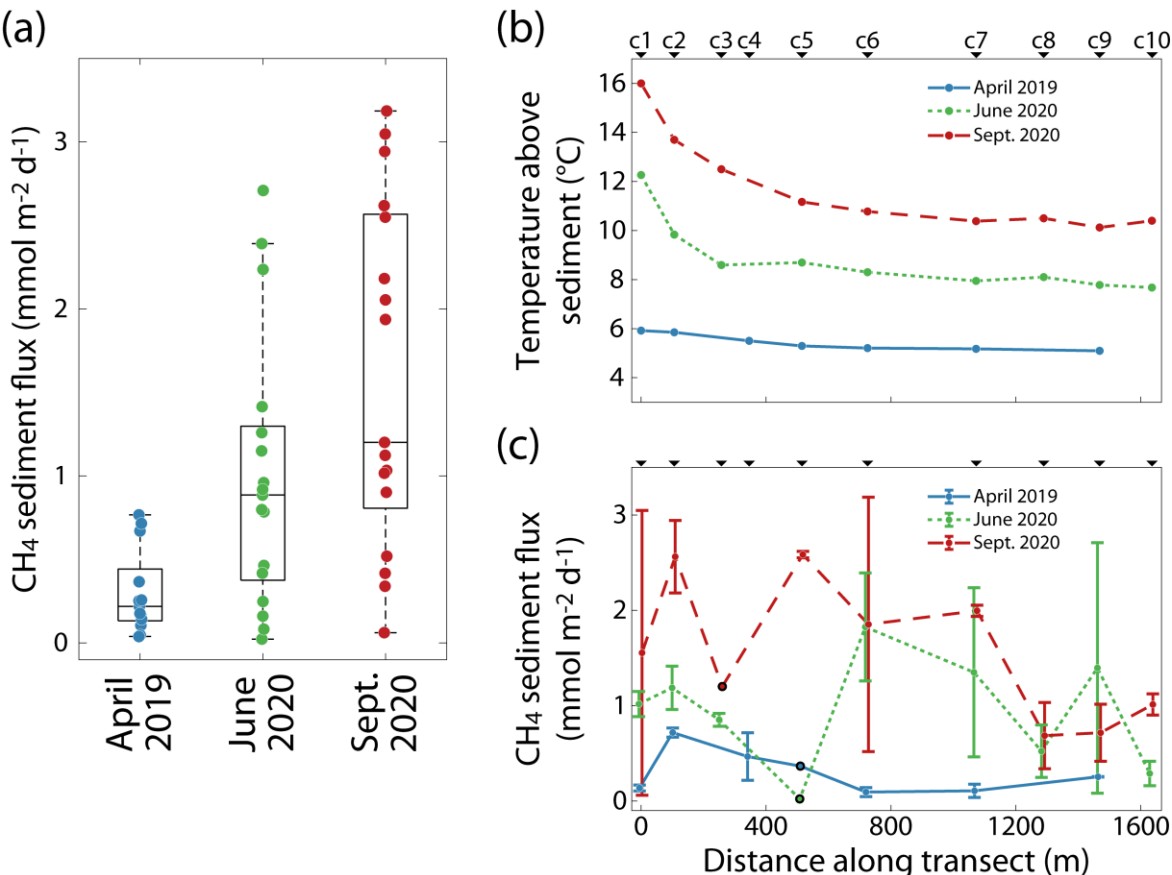

**Figure 5. (a) Diffusive CH₄ flux at the sediment-water interface of each campaign. The middle line of the boxes represent the median, the boxes demarcate the 25th and 75th percentiles, and the whiskers extend to the most extreme data points, not considering outliers. (b-c) Water temperature 0.5 m above the sediment (b) and the mean diffusive CH₄ flux (c) at each station of the transect with either one sediment core (indicated by black contour) or two sediment cores per station at each campaign (Error bars represent the range of values). The individual sampling stations are indicated above the panels.**

## 4 Discussion

### 4.1 Seasonal and spatial differences in pore water CH₄ concentrations

CH₄ pore water concentrations in Schwarzenbach Reservoir ranged between 0.002 mmol L$^{-1}$ and 1.530 mmol L$^{-1}$, which is comparable to what other studies have found in the sediment of freshwater systems (Schulz and Conrad, 1995; Murase and Sugimoto, 2001; Huttunen et al., 2006; Maeck et al., 2013; Norði et al., 2013; Donis et al., 2017). The CH₄ pore water concentrations typically increased with increasing sediment depth (Fig. 3), which is consistent with observations in other studies (Frenzel et al., 1990; Huttunen et al., 2006; Deutzmann et al., 2014). The CH₄ pore water concentrations showed seasonal and spatial differences (Fig. 3). The median CH₄ concentration in the pore water across all measured profiles in September 2020 (0.68 mmol L$^{-1}$) was almost two times larger than in June 2020 (0.37 mmol L$^{-1}$) and about four times larger



than in April 2019 (0.16 mmol L$^{-1}$) (Fig. 3b). The increase in pore water concentration with the season may be explained by
differences in CH$_4$ production within the sediment. Many studies have demonstrated that CH$_4$ production rates in sediments are enhanced at higher temperatures (Schulz et al., 1997; Lofton et al., 2014; Marotta et al., 2014; Shelley et al., 2015; Sepulveda-Jauregui et al., 2018). In Schwarzenbach Reservoir, sediment temperatures increased as the season progressed (Fig. 5b), supporting the hypothesis that higher production was responsible for larger pore water CH$_4$ concentrations later in the season. Enhanced CH$_4$ concentrations were present, especially in deeper sediment layers, which are stronger affected by
changes in production and less affected by CH$_4$ losses due to oxidation and vertical transport than the upper sediment layers. However, the spatial distributions of pore water CH$_4$ along the transect during the different campaigns cannot be explained by temperature alone. Sediment temperatures were elevated in the shallow water zone in June and September 2020, but CH$_4$ concentrations were generally largest near the center of the transect and lowest towards the western and eastern end of the basin (Fig. 3a). Availability of organic matter in the sediment is another factor that correlates with CH$_4$ production (Duc et
al., 2010) or CH$_4$ concentrations (Murase and Sugimoto, 2001) in lake sediments and is considered to be a major limiting factor for CH$_4$ production (Segers, 1998). Settling of organic matter is known to correlate with the current velocity of the inflows (Kufel, 1991). Fewer organic particles might deposit close to the inflows than in the open water because water current velocity is likely to be smaller in the open water of the central basin than near the inflows.

### 4.2 Significance of the potentially released amount of CH$_4$ due to reservoir flushing

Assuming that the CH$_4$ stored in the sediment is completely released into the water during the sediment flushing process and degasses quickly to the atmosphere when the water is rapidly transported out of the reservoir, $N_{CH4,FSL}$ provides a measure of the potential CH$_4$ emission from the reservoir due to flushing. The amount of CH$_4$ mobilized during a flushing operation depends on the depth of sediment erosion and on the season (Fig. 4a). Flushing operation eroding a 15 cm thick sediment layer in the assumed 60 m wide flushing channel along the thalweg of Schwarzenbach Reservoir implies a flushed sediment
volume of $16.5 \times 10^3$ m$^3$, which agrees well with the study by Saam et al. (2019), who assessed the feasibility of reservoir flushing in Schwarzenbach and obtained a flushed sediment volume of $13.6 \times 10^3$ m$^3$ in a simulation of a full drawdown flushing scenario. At an erosion depth of 15 cm, the resulting $N_{CH4,FSL}$ are 6.5 kmol CH$_4$, 8.1 kmol CH$_4$, and 12.5 kmol CH$_4$ if the flushing operation is conducted in April, June, and September, respectively. This suggests that conducting a reservoir flushing operation in spring rather than in late summer would reduce flushing induced CH$_4$ emissions in Schwarzenbach
Reservoir by a factor of 2.

       Typical CH$_4$ emission pathways in Schwarzenbach Reservoir include CH$_4$ emissions due to ebullition (212 mol d$^{-1}$ during normal operation mode), diffusive CH$_4$ emissions (27 mol d$^{-1}$), and CH$_4$ emissions due to degassing during turbination (14 mol d$^{-1}$) (Peeters et al., 2019; Encinas Fernández et al., 2020). Together, CH$_4$ emissions by these pathways add up to 253 mol d$^{-1}$ or 92.4 kmol yr$^{-1}$. Hence, one flushing operation with 15 cm erosion depth potentially causes CH$_4$
emissions that would account for 7 (flushing in April) –14% (flushing in September) of the typical annual CH$_4$ emission in Schwarzenbach Reservoir.



Not only the total amount of $CH_4$ but also the average $CH_4$ concentration in the flushed sediment increases with increasing erosion depth (Fig. 4b). This implies that overall less $CH_4$ is released if an equivalent sediment volume is eroded by several flushings mobilizing thin layers of sediment instead of a few flushings mobilizing thicker sediment layers. For instance, if flushing in April mobilizes the top 5 cm, 10 cm, or 100 cm of sediment, $C_{CH4,FSL}$ is 0.10 mmol $L^{-1}$, 0.27 mmol $L^{-1}$, and 0.74 mmol $L^{-1}$, respectively (Fig. 4b and Fig. S2). Hence, the potential release of $CH_4$ during one flushing event of a 100 cm thick sediment layer is about 2.7 or 7.5, times, larger than 10 flushing events of 10 cm and 20 flushing events of 5 cm thick sediment layers, respectively. Below ~50 cm of sediment, $C_{CH4,FSL}$ is essentially constant (Fig. S2) and thus, $N_{CH4,FSL}$ in flushing events eroding more than 50 cm of sediment increases essentially linearly with the volume of sediment eroded.

Note, that Schwarzenbach Reservoir has been emptied so far only three times since its completion. However, many reservoirs worldwide are flushed regularly, such as the Dashidaira reservoir in Japan (storage capacity of $9 \times 10^6$ $m^3$), where a flushing operation with full water level drawdown is conducted annually (Esmaeili et al., 2017; Sumi et al., 2017). In this reservoir, flushed sediment volumes between $60 \times 10^3$ and $590 \times 10^3$ $m^3$ with an average flushed sediment volume of $287 \times 10^3$ $m^3$ have been reported (Sumi et al., 2009; Esmaeili et al., 2017). Therefore, the average flushed sediment volume in Dashidaira reservoir is 17–21 times larger than the estimated flushed sediment volumes in the study of Saam et al. (2019) and in this study.

Unfortunately, no data is available on $CH_4$ concentration in the sediment or $CH_4$ emissions of Dashidaira reservoir. However, assuming that $CH_4$ concentrations in the sediment of Dashidaira reservoir are comparable to those in Schwarzenbach Reservoir and that the erosion depth is 100 cm or more, the average concentration in the flushed sediment can be approximated by $C_{CH4,FSL}$ of 0.73 mol $m^{-3}$, which is the average $CH_4$ concentrations in the sediment flushed layer at 100 cm erosion depth in Schwazenbach reservoir (Fig. S2). Therefore, the average flushed sediment volume in Dashidaira reservoir of $287 \times 10^3$ $m^3$, would contain 210 kmol $CH_4$. Because the reservoir is flushed each year, the emissions from this reservoir due to flushing would be ~210 kmol $yr^{-1}$, i.e., 2.3 times larger than the typical annual emissions from Schwarzenbach Reservoir. Hence, $CH_4$ emissions due to reservoir flushing operations can contribute substantially to overall $CH_4$ emissions from reservoirs.

$CH_4$ emissions due to reservoir flushing can represent a significant contributing pathway to overall $CH_4$ emissions from reservoirs only if $CH_4$ production in the sediment can provide the amount of $CH_4$ in the flushed sediment volumes between flushing operations. In Schwarzenbach Reservoir, the measured $CH_4$ flux at the sediment-water interface was 1.02 mmol $m^{-2}$ $d^{-1}$ (average over all measurements). Typically, 60–90% of $CH_4$ produced in the sediment is oxidized (Le Mer and Roger, 2001). To compensate the flux from the sediment and the loss due to oxidation of the produced $CH_4$ requires a $CH_4$ production per unit sediment surface of 2.6–10.2 mmol $m^{-2}$ $d^{-1}$. Considering a 100 cm thick layer of sediment, this implies a $CH_4$ production of 2.6–10.2 mmol $m^{-3}$ $d^{-1}$. This estimated production rate is compatible with the median production rate of 10 mmol $m^{-3}$ $d^{-1}$ obtained from direct measurements of production rates in sediments of lakes (the median of a data compilation of 93 samples of different lake sediments; (Martinez-Cruz et al., 2018)). At a $CH_4$ production rate of 2.6–10.2 mmol $m^{-3}$ $d^{-1}$, the time required to produce the average $CH_4$ concentration in the flushed sediment $C_{CH4,FSL} = 0.73$ mol $m^{-3}$ is





73–291 days. This estimation suggests that $CH_4$ production in the sediments provides sufficient $CH_4$ in the flushed sediment layers between annual flushing operations.

**4.3 Additional indirect $CH_4$ emissions due to reservoir flushing**

$CH_4$ emissions due to reservoir flushing might be not only limited to the direct $CH_4$ emissions (i.e., when sediment is eroded and the stored $CH_4$ is released to the water column and eventually into the atmosphere), but a flushing operation might also cause additional indirect $CH_4$ emissions. During a drawdown flushing operation, the water is released through one or more outlets and the water level is decreased dramatically, resulting in an additional drawdown flux, as the stored $CH_4$ in the water column can degas during turbination. Furthermore, as water level fluctuations in reservoirs can enhance $CH_4$ ebullition due to changing hydrostatic pressure (Harrison et al., 2017; Encinas Fernández et al., 2020), lowering the water level during the drawdown period would cause an additional $CH_4$ ebullition flux. While this also reduces the amount of $CH_4$ in the sediment accessible to the emissions due to sediment erosion, the total emission due to reservoir flushing might still be higher, as $CH_4$ production in the sediment could partly replace the $CH_4$ that was lost due to ebullition for the duration of the drawdown period. Lastly, it was shown that the $CH_4$ flux from sediments can be enhanced for several days when they fall dry (Kosten et al., 2018), thus leading to additional $CH_4$ emissions due to exposed sediment areas during the flushing operation.

**5 Conclusions**

Assessing the significance of $CH_4$ emissions due to reservoir flushing in comparison to other pathways requires consideration of the flushing frequency, the flushed sediment volume, the erosion depth during the flushing event, and the vertical and lateral distribution of $CH_4$ concentrations in the sediment. In Schwarzenbach Reservoir, we estimated that one flushing operation could potentially cause $CH_4$ emissions that are comparable to 7–14% of the reservoir's typical annual emissions. Seasonal differences in $CH_4$ concentrations in the sediment suggest that the timing of the flushing operation during the season affects the amount of $CH_4$ emitted. Because the $CH_4$ concentration increases with depth in the sediment, removal of the same sediment volume by regular flushing of shallow sediment layers causes less $CH_4$ emissions than few flushings of thicker sediment layers. In Schwarzenbach Reservoir, flushing is not conducted regularly. However, there are many reservoirs where flushing operations mobilize much larger volumes of sediment and are conducted more regularly than in Schwarzenbach Reservoir. In these reservoirs, $CH_4$ emission due to reservoir flushing is likely substantial, and thus their overall $CH_4$ emission might be severely underestimated.

**Author contribution**

OL: Conceptualization, Formal analysis, Investigation, Project administration, Software, Visualization, Writing – original draft, Writing – review & editing. JEF: Investigation, Project administration, Writing – review & editing. KMC: Resources,



Writing – review & editing. FP: Conceptualization, Formal analysis, Funding acquisition, Project administration, Resources, Software, Supervision, Writing – review & editing.

**Acknowledgements**

We thank Enzo Gronchi, Ramona Ragg, Raymund Hackett, Louis Lauber and Beatrix Rosenberg for technical assistance in the field, and we are grateful to the members of our group for valuable feedback in the early stage of the article.

**Financial Support**

This work was financially supported by the Ministry of Science, Research and the Arts of the Federal State Baden-Württemberg, Germany (Grant: Water Research Network project: Challenges of Reservoir Management – Meeting Environmental and Social Requirements).

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
