# Peer review of "Methane emissions due to reservoir flushing: a significant emission pathway?"

_EGUsphere, 2023_

## Author Comment (AC1)

**Response to RC1**

Below, we respond to the comments made by referee #1. Referee #1's comments are numbered and displayed in black. The author's comments are highlighted in blue. Citations and modifications of the manuscript are *italicized* and quoted.

1. Lessmann et al. presented a very interesting study that reveals the importance of CH4 emissions due to reservoir flushing. To the best of my knowledge, this is one of the first studies that estimate CH4 emissions due to reservoir flushing. Their results indicated that this CH4 emission pathway could be an important missing piece for the reservoir CH4 cycle. Besides, Lessmann et al. also answered several questions that I had in my mind before reading the work, including 1) how this CH4 emission pathway compares with other pathways in this reservoir; 2) how this CH4 emission pathway interacts with other CH4 emission pathways; 3) how important this CH4 emission pathway might be for other reservoirs in the globe. In addition, the work also provides reasonable operation advice to reduce CH4 emissions from this pathway. Although the estimates still have many gaps and the conclusion may not be applied to other reservoirs of different environments, as a pioneer study it will help encourage more following studies to bridge these gaps and address the transferability issue. Overall, I think that it is well-written and all results are clearly explained. I recommend its publication in this journal.

Thank you for taking time to review our work and your positive assessment. We appreciate the helpful comments you provided regarding our manuscript. Please find our response to each comment in detail below.

2. Some comments for the authors to consider. 1) The manuscript does not provide the information of reservoir age. Previous studies showed that the transition of carbon dynamics with reservoir aging is significant (Maavara et al., 2020). It is thus valuable that the authors can put their estimates and discussion in this context and warn the audience that the importance of this pathway can change significantly in time. Maavara, T., Chen, Q., Van Meter, K., Brown, L. E., Zhang, J., Ni, J., & Zarfl, C. (2020). River dam impacts on biogeochemical cycling. Nature Reviews Earth & Environment, 1(2), 103-116.

Thank you for pointing out this study. Maavara et al. discuss that greenhouse gas (GHG) emissions are initially high in young reservoirs due to flooding and decomposition of terrestrial biomass, are decreasing in middle-aged reservoirs with decreasing flooded biomass, and are increasing again in old reservoirs with sediment accumulation (2020). Because reservoir flushing affects the sediment budget, the accumulation of sediment with reservoir age does not necessarily follow the typical processes as outlined in the suggested paper. Thus, the relevance of CH$_4$ release due to flushing concerning reservoir age might be very complicated and is beyond the scope of our study. Nevertheless, we have included the information about the reservoir's age in section 2.1:

*"Since the dam was completed in 1926, the reservoir has been completely emptied on three occasions in 1935, 1952, and 1997."*

3. 2) The method to estimate diffusive CH4 flux from sediment to the water column is only accurate when there aren't any large CH4 production or oxidation in the surface sediment layers. But Figure 3d shows that the gradient of CH4 concentration changes between 0 and 2.5 cm depth, implying that CH4 oxidation occurred during April 2019 sampling and CH4 production occurred

during both June and September 2020 sampling. Do the authors have a sense of the related estimate uncertainty?

We calculated the instantaneous diffusive sediment flux of $CH_4$, by multiplying the actual gradient of the $CH_4$ pore water concentration by the molecular diffusivity of $CH_4$ in water. This does not require a steady-state assumption. Production and oxidation of $CH_4$ may alter the gradient over time, but not our estimate of the instantaneous diffusive sediment flux.

One difficulty comes from the limitation of the experimental technique. We estimate the gradient in the $CH_4$ concentration by linear regression within the uppermost sediment layers (in our case: 0.25–2.25 cm). Hence, the estimated diffusive flux is calculated from the average gradient within these sediment layers, which is not the true local gradient across the sediment-water interface. If substantial oxidation occurs above our uppermost sample, the sediment flux into the water is smaller than our estimate. However, this does not affect our estimate of the minimum production required to generate the estimated fluxes.

Another problem is that if the $CH_4$ pore water concentrations change non-linearly within the uppermost sediment layers, it would imply that the diffusive fluxes vary within these sediment layers. In that case, our estimate is always lower than the maximum sediment flux in these sediment layers.

4.  3) Is the y-axis of Figure 2b really sediment depth? I suspect it is still water depth, consistent with Figure 2a. Anyway, I cannot tell that DO in September 2020 were oversaturated near the water surface and above 2 mg throughout the entire water column from the figure, as described in Line 180.

We apologize for this mistake. The y-axis in Figure 2b should be labeled *"Height above ground (m)"*, consistent with Figure 2a. We changed it accordingly.

---

## Author Comment (AC2)

**Response to RC2**

Below, we respond to the comments made by referee #2. Referee #2's comments are numbered and displayed in black. The author's comments are highlighted in blue. Citations and modifications of the manuscript are *italicized* and quoted.

1. This study measures vertical profiles of methane in reservoir sediment porewater to estimate the potential for enhanced methane emission during sediment flushing events (a management strategy employed to reduce sediment accumulation in reservoirs). The authors then compare the magnitude of these potential emissions to existing estimates of methane emission from the reservoir via more commonly recognized pathways. The study is the first that I know of to consider the potential methane emission impacts of a management practice that is likely quite common in certain types of reservoir ecosystems (I was previously aware of this practice particularly for reservoirs in tropical/monsoon driven climates). While the study is limited in it's reach (e.g. it doesn't document emissions during a flushing event; it just estimates potential), it points out an important gap in our understanding and the findings suggest that this emission pathway should be studied in more detail.

First, thank you for taking the time to review our work. We appreciate your comments and suggestions, which helped us improve this manuscript. Below, you'll find our response to each comment in detail.

2. I have two major comments: 1.) I think the authors should dial back their conclusions a bit and place more emphasis on the need for empirical study of these flushing events. I suggest adding a bit more text discussing the potential for methanotrophy once sediment is mobilized and how this may depend on travel time to the dam outlet. 2.) To the extent possible, the authors should spend a bit more time familiarizing the readership with flushing events as a management practice (for example: do we have any idea how common a practice is this? how well constrained are estimates of sediment erosion during these events (both total volume and spatial/depth relationships?). I think a table describing some of this literature would be helpful for putting this in context and helping the readership to understand the potential importance. Particularly, the information about flushing frequency and all the papers cited on lines 63-65. How different are the reservoirs where flushing is conducted (e.g. do they generally have to be small enough for the sediment plume to have a relatively short residence time?)
Overall I think this is a creative and timely paper that points to important areas for future work with potentially important management implications.

1) We made changes in different sections of the manuscript to emphasize the need for further research (see our responses to comment no. 6, 30 and 32).

We acknowledge the importance to address the role of $CH_4$ oxidation during sediment flushing events. Unfortunately, we did not measure any methanotrophic rate as it was out of our scope. Nevertheless, we added a small discussion regarding $CH_4$ oxidation at the end of section 4.2:

*"The total $CH_4$ emissions during a flushing event may be reduced by $CH_4$ oxidation. However, the free-flow state during drawdown flushing, when most of the $CH_4$ is mobilized, is typically maintained only for several hours, depending on the desired amount of flushed sediment (Kondolf et al., 2014).*

*Therefore, given the relatively short residence time of the CH$_4$-rich water, the potential for oxidation of the mobilized CH$_4$ is rather limited."*

2) Initially, our idea was to compile a list of reservoirs with their respective total flushed sediment volumes and base our analysis on these. However, we found that information regarding total flushed sediment volume as well as erosion depth and area was scarcely reported. Moreover, many studies reported total flushed sediment in tons, which would have resulted in additional uncertainties when converting into sediment volume. The only parameter that was reported relatively consistent across studies was the flushing frequency, which we reported in Lines 62–65.

To give readers a better insight into reservoir flushing, we explained drawdown flushing in more detail, based on the work of Kondolf et al. (2014). We also added information about the diversity and characteristics of reservoirs with flushing operations. Furthermore, we improved the readability of this section by removing one sentence with repeating information (compare Lines 60-70). The section now reads as follows (starting at Line 56):

*"Deposited sediment can be removed by mechanical removal (e.g., dredging) or hydraulic flushing, which utilizes the eroding force of water currents. Drawdown flushing represents a typical form of hydraulic flushing, that consists of three basic steps: completely drawing down the water level, maintaining a free-flow state, and recovering of water levels (Kondolf et al., 2014). During the free-flow state, sediment is mobilized and flushed through a dam outlet to the downstream river section. Consequently, any CH$_4$ previously stored in the sediment pore water is also mobilized into the water stream and can eventually degas to the atmosphere at the reservoir surface, during turbination, or downstream of the reservoir, leading to CH$_4$ emissions. Sediment flushing operations have been conducted worldwide in reservoirs of all sizes, with (initial) storage capacities widely ranging from 0.8 to 9640 × 10$^6$ m$^3$ (Sumi et al., 2017; Antoine et al., 2020). Among reservoirs with relatively regular flushing operations, flushing frequencies between twice a year and once every five years have been reported (Brandt and Swenning, 1999; Chang et al., 2003; Kantoush and Sumi, 2010; Fruchard and Camenen, 2012; Grimardias et al., 2017; Sumi et al., 2017; Antoine et al., 2020), but there are also reservoirs that are flushed irregularly (Sumi et al., 2017). Reservoirs with higher sediment yield need to be flushed more frequently than others. For instance, the Tapu Reservoir in Taiwan is characterized by high sedimentation rates induced by heavy rainfalls and steep-sloped mountains and was flushed ten times between 1991 and 1996 (Chang et al., 2003). The total amount of flushed sediment depends on reservoir geometry, sediment characteristics and operation strategy (Morris, 2020; Petkovšek et al., 2020) and the conditions for a successful flushing operation are typically optimal in smaller reservoirs that are long and narrow (Kondolf et al., 2014)."*

Reference:

Kondolf, G. M., Gao, Y., Annandale, G. W., Morris, G. L., Jiang, E., Zhang, J., Cao, Y., Carling, P., Fu, K., Guo, Q., Hotchkiss, R., Peteuil, C., Sumi, T., Wang, H.-W., Wang, Z., Wei, Z., Wu, B., Wu, C., and Yang, C. T.: Sustainable sediment management in reservoirs and regulated rivers: Experiences from five continents, Earth's Futur., 2, 256–280, https://doi.org/10.1002/2013EF000184, 2014.

3. Line by Line:
   Line 6: consider rephrasing: "Reservoirs are globally significant sources of the greenhouse gas methane"

We rephrased the sentence as follows:

*"Reservoirs represent a globally significant source of the greenhouse gas methane (CH₄), which is emitted via different emission pathways."*

4. Line 16: add "seasonal" between "depends on the" and "timing"

We added the word *"seasonal"* accordingly. With additional changes based on comment no. 25, the sentence now reads as follows:

*"CH₄ emissions due to one flushing operation can constitute 7–14% of the typical annual CH₄ emissions from Schwarzenbach Reservoir, whereby the amount of released CH₄ depends on the seasonal timing of the flushing operation and can differ by a factor of two."*

5. Lines 16-17: Another potentially more straightforward way to say this would be: "Larger flushing events that mobilize deeper sediments lead to non-linear increases in CH4 mobilization"

We rephrased the sentence as follows:

*"Larger flushing events that mobilize deeper sediment layers lead to non-linear increases in CH₄ mobilization."*

6. Lines 21-22: I think I'd use this sentence to describe important next research steps—we need to know a lot more before we can start including this pathway in large scale estimates.

We rephrased the last sentence of the abstract as follows:

*"Our study recognizes CH₄ emissions due to reservoir flushing as an important pathway, identifies potential management strategies to mitigate these CH₄ emissions, and emphasizes the need for further research."*

7. Line 24: I'm not a fan of stating the number of global reservoirs this precisely… it makes it sound like there isn't much uncertainty in this number (even thought there is). Consider "Worldwide millions of reservoirs have been constructed"—and possibly add a reference to Couto and Olden 2018 (https://doi.org/10.1002/fee.1746).

We added the reference and changed the sentence as suggested:

*"Worldwide millions of reservoirs have been constructed (Lehner et al., 2011; Couto and Olden, 2018), and […]"*

8. Line 28: omit "the required"

We removed *"the required"*. The sentence now read as:

*"Today, we know that reservoirs represent a significant source of GHG emissions, […]"*

9. Line 30: Check units on this reported range (your high number looks too high). See a new review by Lauerwald et al. 2023 (https://doi.org/10.1029/2022GB007657) for a range of 10-52 Tg CH4 yr-1 from existing global reservoir estimates (which includes the lower estimate by Johnson et al. 2021). I also suggest you include yr-1 in your units (rather than just saying "annual" budget)

Thank you for bringing this new review to our attention. We changed the sentence and are now using the suggested reference as the only reference, as it is the most recent and includes previous estimates for global $CH_4$ emissions:

*"[...] and researchers have estimated that reservoirs contribute around 10–52 Tg $CH_4$ $yr^{-1}$ to the global budget of atmospheric $CH_4$ (Lauerwald et al., 2023)."*

10. Lines 33, 42 and throughout: I suggest avoiding the term "drawdown" flux as synonymous with degassing flux. When I hear the term "drawdown" flux I think of either: 1.) the ebullition flux that occurs when hydrostatic pressure drops during reservoir drawdown or 2.) the flux from drying littoral sediments that occurs when water levels are drawn down

We agree that the term *"drawdown flux"* can be confusing, especially as we are often describing the drawdown of the water level. We removed the term throughout the manuscript or described it as degassing during turbination. The affected sentences read now as follows:

i) *"$CH_4$ can be emitted from reservoirs via different pathways such as ebullition, plant-mediated transport, diffusion across the water-atmosphere interface, degassing during turbination and [...]"*

ii) *"While most of these emission pathways usually exist in both lakes and reservoirs, the degassing of $CH_4$ during turbination, occurs only in reservoirs [...]"*

iii) *"During a drawdown flushing operation, the water is released through one or more outlets and the water level is decreased dramatically, resulting in additional $CH_4$ emissions, as the $CH_4$ stored in the released water can degas during turbination and downstream of the reservoir."*

11. Line 41: I suggest referencing Denfeld et al 2018 (https://doi.org/10.1002/lol2.10079) here and in the discussion. I think it would be helpful to discuss the importance of constraining how much of the flushed sediment methane might get oxidized before passing downstream. You could draw from the turnover literature for this, but also discuss the importance of the sediment plume residence time.

Thank you for this suggestion. We addressed the importance of $CH_4$ oxidation during the flushing event in the discussion section 4.2 (see comment no. 2). We also included another reference in Lines 39–40:

*"Intensive vertical mixing during spring and fall overturn causes rapid transport of the stored $CH_4$ to the water surface from where it diffuses to the atmosphere (Bastviken et al., 2004)."*

Reference:

Bastviken, D., Cole, J., Pace, M., and Tranvik, L.: Methane emissions from lakes: Dependence of lake characteristics, two regional assessments, and a global estimate, Global Biogeochem. Cycles, 18, 1–12, https://doi.org/10.1029/2004GB002238, 2004.

12. Line 45: I suggest citing Harrison et al. 2021 (https://doi.org/10.1029/2020GB006888) who estimated that over half of the global methane emissions from reservoirs may occur via degassing.

Thank you for the suggestion. We added the reference:

*"[…] where it can become the dominant source of $CH_4$ emissions (Kemenes et al., 2007; Harrison et al., 2021) […]"*

13. Line 49: Or load following/hydropeaking operations & ship lock induced changes in water level (Maeck et al 2014 https://doi.org/10.5194/bg-11-2925-2014; Harrison et al. 2017)

We added the information regarding hydropeaking and ship lock operations as part of the regularly occurring decrease in pressure and added more references:

*"Therefore, water level drawdowns in reservoirs can substantially increase the ebullition flux of $CH_4$ due to decreasing pressure, which can become particularly large during fall drawdown or occur regularly during diel pumped-storage operations, as well as hydropeaking and ship lock operations (Maeck et al., 2014; Harrison et al., 2017; Almeida et al., 2020; Encinas Fernández et al., 2020)."*

Reference:

Almeida, R. M., Hamilton, S. K., Rosi, E. J., Barros, N., Doria, C. R. C., Flecker, A. S., Fleischmann, A. S., Reisinger, A. J., and Roland, F.: Hydropeaking Operations of Two Run-of-River Mega-Dams Alter Downstream Hydrology of the Largest Amazon Tributary, https://doi.org/10.3389/fenvs.2020.00120, 2020.

14. Line 115: This is the first paper where I've seen NaCl used to reduce the solubility of methane in the water sample (and force it into the headspace). I see there is another published paper using this approach, but it isn't common to my knowledge. According to my bunsen's solubility calculations, the concentration of NaCl you used would reduce solubility by about an order of magnitude (from 0.03 to 0.0015 liters per liter at STP). I might just add a line explaining whether you calculated the methane still dissolved in solution (or whether this was just assumed to be nominal?)

The provided calculations match our values, but we are not sure if Bunsen's solubility is valid for salinities of the magnitude in our samples (222–375 g $L^{-1}$). The technique was previously used by Hofmann et al. who describe that nearly 100 % of the CH4 degasses into the headspace (2010).

Reference:

Hofmann, H., Federwisch, L., and Peeters, F.: Wave-induced release of methane: Littoral zones as a source of methane in lakes, Limnol. Oceanogr., 55, 1990–2000, https://doi.org/10.4319/lo.2010.55.5.1990, 2010.

15. Line 154: Looking at the rest of the paper, I'm unclear what the C CH4sed estimates deeper than 15 cm were used for? The estimates provided were all using the 1-15cm layer (unless I missed something)? If you do use these deeper concentration estimates, then I suggest bounding them (since the April profile suggests that methane concentration likely continues to increase with depth past 15 cm).

We extended the $C_{CH4,sed}$ to a sediment depth of 100 cm to be able to estimate the average concentration of $CH_4$ in the sediment flushing volume ($C_{CH4,FSL}$) for sediment layers below 15 cm. Specifically the $C_{CH4,FSL}$ at 100 cm sediment depth was used for (i) discussing the differences in $CH_4$ mobilization when flushing an equivalent sediment volume by flushings with a different frequency–volume ratio (Line 277–284), (ii) estimating the amount of $CH_4$ in the average flushed sediment volume in Dashidaira Reservoir (Line 293–297), and (iii) comparing $C_{CH4,FSL}$ to the $CH_4$ production rate to estimate the time required to produce the average $CH_4$ concentration in the flushed sediment (Line 305–312).

Note that in the first example (i) we used the $C_{CH4,FSL}$ at 100 cm sediment depth for April, to compare $CH_4$ mobilization within the same seasonal time, while in the other two examples (ii–iii) we used the $C_{CH4,FSL}$ at 100 cm sediment depth, averaged over all seasonal profiles.

To clarify this, we added the description *"seasonal average"*. The section reads as follows:

*"However, assuming that $CH_4$ concentrations in the sediment of Dashidaira Reservoir are comparable to those in Schwarzenbach Reservoir and that the erosion depth is 100 cm or more, the average concentration in the flushed sediment can be approximated by $C_{CH4,FSL}$ of 0.73 mol $m^{-3}$, which is the average $CH_4$ concentrations (seasonal average) in the sediment flushed layer at 100 cm erosion depth in Schwarzenbach Reservoir (Fig. S2)."*

16. Line 174: Cite the papers that you are using for CH4 emissions via other pathways at Schwarzenbach here.

We added the references as suggested.

17. Figure 2: I'm confused by your y-axes. A standard approach in limnology is to plot depth on the y axis (with values in reverse order where 0m is at the top and 40m is at the bottom). Also, I think you are plotting water column dissolved oxygen concentrations in panel B, but the y axis suggests these may be porewater concentrations?

We apologize for this mistake. The y-axis in Figure 2b should be labeled *"Height above ground (m)"*, consistent with Figure 2a.

We preferred to work with the height above ground, as we use parameters for our calculations that are measured just above the sediment (e.g., temperature), as well as to better represent the large differences in water level (the water level in April was about 3 m higher than in June and September).

18. Line 210-211: Is there much literature on different erosion depths? If so, then explain your assumption. If not, then argue for more study of this in the future.

The assumption for the width of the flushing channel (60 m) was based on a study in Dashidaira Reservoir, a reservoir similar in size that has been studied intensively in terms of flushing operations. For the sediment depth, we were limited to 15 cm as the deepest common measurement of $CH_4$ across all 47 sediment cores was at 15 cm sediment depth. At 15 cm erosion depth the resulting flushed sediment volume was comparable to the reported volume (obtained by simulation of a drawdown flushing scenario in Schwarzenbach Reservoir) of another study (Saam et al., 2019) (see Line 263–267). Thus, we considered our assumptions fair for a first estimation of potential $CH_4$ emissions due to reservoir flushing.

Furthermore, to the best of our knowledge, most of the studies that work in the field of reservoir flushing, do not consistently report erosion depths. We found two studies examining reservoir flushing events in Dashidaira Reservoir, from which we could obtain erosion depths. In one, erosion depths between 67 and 203 cm have been reported (Esmaeili et al., 2017); in the other, typical erosion depths around 100–200 cm can be deduced from the reported figures (Esmaeili et al., 2015). Although the size and capacity of both reservoirs (Schwarzenbach and Dashidaira) are comparable, erosion depth may range significantly depending on the sediment's characteristics and basin morphology.

We rephrased the information regarding our assumptions in section 2.5 as follows:

*"Assumptions for flushing channel width (60 m) were based on reported values for Dashidaira Reservoir, a reservoir of similar size (Storage capacity: $9 \times 10^6$ $m^3$; Surface area: 0.35 $km^2$) that has been extensively studied in terms of flushing operations (Kantoush et al., 2010; Esmaeili et al., 2015; Esmaeili et al., 2017). Furthermore, the flushing channel was assumed to extend basin-wide along the thalweg of Schwarzenbach Reservoir (Fig. 1a)."*

We also modified the sentence in Line 210–211 to be consistent with the description of the same result in Lines 267–268 as follows:

*"At an erosion depth of 15 cm, $N_{CH4,FSL}$ was 6.5, 8.1 and 12.5 kmol $CH_4$ in April 2019, June 2020 and September 2020, respectively."*

References:

Saam, L., Mouris, K., Wieprecht, S., and Haun, S.: Three-dimensional numerical modelling of reservoir flushing to obtain long-term sediment equilibrium, in: Proceedings of the 38th IAHR World Congress (Panama), Panama City, Panama, 1–6, https://doi.org/10.3850/38WC092019-0742, 2019.

Esmaeili, T., Sumi, T., Kantoush, S. A., Kubota, Y., Haun, S., and Rüther, N.: Three-dimensional numerical study of free-flow sediment flushing to increase the flushing efficiency: a case-study reservoir in Japan, Water, 9, 900, 10.3390/w9110900, 2017.

Esmaeili, T., Sumi, T., Kantoush, S. A., Kubota, Y., and Haun, S.: Numerical study on flushing channel evolution, case study of Dashidaira reservoir, Kurobe river, Journal of Japan Society of Civil Engineers, Ser. B1, 71, I_115-I_120, https://doi.org/10.2208/jscejhe.71.I_115, 2015.

19. Figure 3d and Figure 4b seem very similar to me. Do you need 4b?

Figure 3d shows the median $CH_4$ concentration in each (individual) sediment layer across all measured profiles. Figure 4b (and Supplementary Figure S2) shows the average $CH_4$ concentration as a function of flushed sediment layer thickness, which is based on the estimated flushing volume in Schwarzenbach Reservoir. We need this information to calculate the amount of $CH_4$ in a flushed sediment volume with specific erosion depth (e.g., as discussed throughout section 4.2).

20. Line 239: Deemer and Harrison 2019 (https://doi.org/10.1007/s10021-019-00362-0) is another study that contains sediment porewater methane concentrations for a small reservoir—maybe not necessary to add, but it does also discuss how timing of drawdowns (relative to seasonal methane accumulation) can be important for determining water column flux from sediment.

Thank you for bringing this recent study to our attention. We added it as a reference for $CH_4$ porewater concentrations, as we lack references for $CH_4$ porewater concentrations in reservoirs.

21. Line 246: add "relative to methane oxidation rates" after "are enhanced at higher temperatures". Also, the Shelley paper you reference showed that oxidation may keep up with methane production as temperatures warm in some systems, so you could do a "but see" reference there.

Thank you for the suggestion, which we added as follows:

*"Many studies have demonstrated that $CH_4$ production rates in sediments are enhanced at higher temperatures relative to $CH_4$ oxidation rates (Schulz et al., 1997; Lofton et al., 2014; Marotta et al.,*

*2014; Sepulveda-Jauregui et al., 2018), although, in some systems, $CH_4$ oxidation rates may keep up with $CH_4$ production rates (Shelley et al., 2015)."*

22. Line 254:  Might spatial variability in bottom water dissolved oxygen concentration also explain this?

Throughout our study, the whole water column was always fully oxygenated (as seen in the dissolved oxygen profiles in Figure 2b). Therefore, we would not expect any changes in $CH_4$ production due to anoxic conditions.

23. Line 261 (and elsewhere):  How rapidly might this transport happen?

We added more information on this topic in the introduction when we explain drawdown flushing (see comment no. 2).

24. Line 265-267:  I think you should mention this scoping study way earlier (when you introduce the study site).  As I read the paper I was wondering if flushing was actually ever conducted at Schwarzenbach until I got to this line in the paper.

We added the following information when describing the study site in section 2.1:

*"Since the dam was completed in 1926, the reservoir has been completely emptied on three occasions in 1935, 1952, and 1997."*

25. Line 268-270:  Cool insight?  Could you work this factor of 2 estimate into the abstract?

Thank you for this suggestion. We implemented this insight into the abstract:

*"$CH_4$ emissions due to one flushing operation can constitute 7–14% of the typical annual $CH_4$ emissions from Schwarzenbach Reservoir, whereby the amount of released $CH_4$ depends on the seasonal timing of the flushing operation and can differ by a factor of two."*

26. Lines 285-291:  Make sure to tell the reader that the reservoirs are similar in size-- I had to look back in the study site description to see that the volumes were similar.  Similarly on line 298 you could provide an areal rate in addition to a total mass.

We added the information that the reservoirs are similar in size:

*"Note, that Schwarzenbach Reservoir has been emptied so far only three times since its completion. However, many reservoirs worldwide are flushed regularly, such as the Dashidaira Reservoir in Japan, a reservoir of similar size, where a flushing operation with full water level drawdown is conducted annually (Esmaeili et al., 2017; Sumi et al., 2017)."*

Instead, we removed the information about the storage capacity of Dashidaira Reservoir in this section, as the storage capacity as well as the surface area is now mentioned earlier in section 2.5 (see also comment no. 18).

Additionally, we provided the amount of $CH_4$ per surface area for Dashidaira Reservoir and compared these to the respective values for Schwarzenbach Reservoir:

*"Because the reservoir is flushed each year, the emissions from this reservoir due to flushing would be 210 kmol $yr^{-1}$ or, with consideration of the reservoir's surface area, 0.6 mol $m^{-2}$ $yr^{-1}$., i.e., 2.3, or 4.3 times larger than the typical annual emissions from Schwarzenbach Reservoir, respectively."*

27. Line 302: change "provide" to "replenish"

We changed the word.

28. Line 310: you could point out that theory would suggest replenishment rates would be higher in more eutrophic reservoirs

$CH_4$ production rates in eutrophic systems are higher than in oligotrophic systems. Thus, the amount of $CH_4$ in the sediment, accessible to sediment erosion, would be higher as well. Consequently, replenishment rates might be similar between systems. An assessment would also require $CH_4$ oxidation rates in different systems, which was out of the scope of this study.

29. Lines 314-315: suggest rephrasing for clarity—"In addition to methane emissions driven by sediment erosion, the water level drawdowns that accompany reservoir flushing events may also lead to enhanced methane emissions."

Thank you for this suggestion. The sentence now reads as follows:

*"In addition to $CH_4$ emissions driven by sediment erosion, the water level drawdown that is necessary during reservoir flushing operations may also lead to enhanced $CH_4$ emissions."*

30. Line 321-324: This sentence is hard to digest. I suggest rephrasing to something that emphasizes uncertainty a bit more… maybe something like "The relative role of erosive forces versus hydrostatic pressure drops in driving methane emissions during reservoir flushing events is an area for future work."

We agree that this sentence was difficult to understand. We tried to break down the information in this section for easier reading, considering your suggestions:

*"Furthermore, as water level fluctuations in reservoirs can enhance $CH_4$ ebullition due to changing hydrostatic pressure (Harrison et al., 2017; Encinas Fernández et al., 2020), lowering the water level during the drawdown period would cause an additional $CH_4$ ebullition flux. If the ebullition originates from the sediment that is mobilized during flushing, the total $CH_4$ emissions during the entire operation do not differ substantially from the above estimates of $CH_4$ release during sediment erosion. However, $CH_4$ ebullition may originate from sediment layers deeper than the eroded sediment. As a result, total $CH_4$ emissions might be larger than the $CH_4$ emissions estimated by sediment erosion alone. The relative contribution of each pathway to the overall $CH_4$ emissions due to reservoir flushing may be an area for future studies."*

31. Lines 324-325: Enhanced methane emissions from drying sediments is not a ubiquitous pattern & some studies show dried sediments can act as a methane sink (Yang et al. 2012; https://doi.org/10.1029/2011JD017362). I suggest rephrasing that this is an important area for more research.

Thank you for pointing out this study. We added the reference and rephrased the sentence as follows:

*"Lastly, throughout a flushing operation, large sediment areas may fall dry and become inundated again. Drying and rewetting of sediments are known to affect $CH_4$ emissions and is an important area for more research (Yang et al., 2012; Kosten et al., 2018)."*

32. Line 332- change "causes" to "will likely cause". Your study is estimating emission potential, but you don't know how methanotrophy will play into ultimate emission dynamics. I think you should be sure to emphasize that emissions should be studied during an actual reservoir flushing event to learn more.

We changed the wording accordingly. Furthermore, we added the following sentence to emphasize the need for future research at the end of the conclusions section:

*"Our findings highlight $CH_4$ emissions due to reservoir flushing as an important area for future research and emphasize the need for field studies during reservoir flushing operations to better understand underlying processes."*

33. Line 336: delete "might be severely"—I wouldn't call the values you present "severe", but you can state that ignoring the pathway leads to underestimation.

We changed the sentence accordingly:

*"In these reservoirs, $CH_4$ emission due to reservoir flushing is likely substantial, and thus their overall $CH_4$ emission will be underestimated if this pathway is ignored."*